# Optimizing brinjal (*Solanum melongena* L.) health and yield through bio-organic amendments against Fusarium wilt

**Jannatul Mauya[1], Khadija Akhter[1], Hasiba Kabir Mim[1], Marzia Rahman[1], Raihan Ferdous** [2]*

**1** Department of Plant Pathology, Sher-e-Bangla Agricultural University, Dhaka, Bangladesh, **2** Mushroom Development Institute, Savar, Dhaka, Bangladesh

* raihanf.agri-2011106@sau.edu.bd

## Abstract

A field experiment was conducted to evaluate the effect of different bio-organic treatments, including control ($T_0$), spent mushroom substrate-SMS ($T_1$), vermicompost ($T_2$), poultry manure ($T_3$), biochar ($T_4$), SMS with biochar ($T_5$), SMS with poultry manure ($T_6$), and SMS with vermicompost ($T_7$) on the management of Fusarium wilt of brinjal (*Solanum melongena* L.). This experiment was followed by a Randomized Complete Block Design (RCBD) using three replications. Pathogen identification confirmed *Fusarium* sp. as the causal agent of wilt through morphological and pathogenicity tests. Growth attributes, yield, and disease incidence were recorded at various growth stages. The highest disease incidence was observed in the untreated control ($T_0$), with values of 44.4%, 55.55%, and 77.77% at 25, 45, and 65 days after transplanting (DAT), respectively. In contrast, the lowest incidence was recorded in $T_5$ (5.55%, 16.66%, and 16.66% at 25, 45, and 65 DAT), followed by $T_6$, with both treatments showing statistically similar results. These findings were statistically significant compared to $T_0$. The treatment of $T_5$ significantly enhanced plant growth parameters, including plant height, number of leaves, and number of branches, alongside a substantial increase in yield (12.71 tons/ha), which was statistically similar to $T_6$. These bio-organic amendments not only suppressed disease incidence effectively but also improved soil health, enhanced microbial diversity, and promoted vegetative growth and yield. The results indicate that the integration of SMS with Biochar or Poultry Manure is a sustainable and eco-friendly strategy for managing Fusarium wilt in brinjal cultivation, potentially replacing conventional chemical methods for enhanced productivity and soil health.

## Introduction

Brinjal (*Solanum melongena* L.) is the most popular, widely available, and reasonably priced vegetable that belongs to the family Solanaceae. In Bangladesh, "Begoon"

**Data availability statement:** All relevant data are within the manuscript.

**Funding:** The author(s) received no specific funding for this work.

**Competing interests:** The authors have declared that no competing interests exist.

(also known as brinjal or eggplant) is a common and beloved vegetable that has a connection to the social, cultural, and rural residents' financial circumstances. In the Kharif (summer) season in Bangladesh, 49406.92 acres were cultivated, with total production at 209541.17 MT, and in the Rabi (winter) season, 84536.65 acres were cultivated, with total production 409001.46 MT [1]. The fruit of the brinjal is very low in calories and has a healthy amount of minerals. Brinjals are rich in anthocyanin chemicals. The strong protective effects of anthocyanins against diabetes, cancer, cardiovascular disease. [2]. Diseases caused by fungi, bacteria, viruses, and nematodes severely reduce the amount of brinjal produced [3]. Wilt of brinjal (*Solanum melongena*) is one of the most damaging diseases affecting brinjal production and can cause substantial economic losses. Several *Fusarium* species have been reported to infect brinjal; however, the most important and devastating among them is *Fusarium oxysporum* f. sp. *melongenae*, the causal agent of Fusarium wilt. This disease significantly reduces both the yield and quality of brinjal, with severity ranging from 10% to 90% [4]. Fusarium wilting generally causes 20–30% of eggplant plants to die, and it may turn into an epidemic during November to December [5]. *Fusarium oxysporum* can survive in the soil with its chlamydospores for a long period of time without any host *Fusarium oxysporum* is a soil-borne pathogen that makes it challenging to regulate. Several control measures, such as crop rotation, the use of various resistant cultivars, soil sterilization and solarization, and the application of fungicides, are some useful or effective management strategies for Fusarium wilt. However, due to the possible negative impacts of fungicides on the environment and human health as well as their unfavorable effects on nontarget organisms, the extensive use of chemical fungicides has been a source of public worry and security [6]. Biological control could be a successful alternative to chemicals. Several organic byproducts, such as spent mushroom substrate (SMS), poultry manure, and vermicompost, have the ability to manage some soil-borne pathogens. Because of its high cation exchange capacity (CEC), slow mineralization rate, and rich nutritional status, the SMS has been discovered to be an excellent source of nutrients for agriculture [7]. SMS is rich in diverse microorganisms, such as disease-antagonistic bacteria and fungi. It is biodegradable, safe to apply, and less expensive to develop [8]. Vermicomposting is the natural process of rotting or decomposition of organic matter by the activity of earthworms under controlled conditions [9]. Vermicompost's organic carbon slows the release of nutrients into the system so that they can be absorbed by the plant. [10]. Poultry manure is an effective soil amendment that supplies nutrients for growing crops. Poultry manure has a high nutritional value since it contains nitrogen, phosphorus, and potassium. [11]. The incidence of wilt (*F. oxysporum*) can be reduced 41.15% by poultry manure [12]. Biochar is a type of charcoal that is produced through a process called pyrolysis, which involves heating organic materials (such as wood chips, agricultural waste, or animal manure) in the absence of oxygen. Biochar acts as a biocontrol agent against various plant pathogens and insect pests by modulating plant physiological functions (such as the activation of defense-related genes), altering rhizosphere chemistry, enhancing beneficial microbial communities, and releasing allelopathic compounds that suppress pathogen development [13]. As

a result, it has been utilized to boost agricultural productivity, enhance soil health, and lower greenhouse gas emissions [14]. This study proposes the use of spent mushroom substrate to combat the pathogen *Fusarium oxysporum* and to find out the effectiveness of Spent Mushroom Substrate and other soil amendments for the management of wilt of brinjal (*Fusarium oxysporum*).

## Materials and methods

### Study site and period

The field experiment was conducted in the Central Farm of Sher-e-Bangla Agricultural University, and the lab experiment was conducted in the MS laboratory of the Department of Plant Pathology, Sher-e-Bangla Agricultural University, Dhaka-1207. The experiment was carried out during the period from September 2021 to April 2022.

### Selection of varieties and treatments

Brinjal variety BARI BT Brinjal 2 (Kajla) was used for the experiment and the variety was collected from Bangladesh Agricultural Development Corporation (BADC), Gabtoli, Dhaka. Along with control, seven treatments were selected and evaluated through a field experiment in natural conditions (*in vivo*). The treatments used in this study are as follows:

$T_0$ = Control,

$T_1$ = Spent Mushroom Substrate (SMS),

$T_2$ = Vermicompost,

$T_3$ = Poultry manure,

$T_4$ = Biochar,

$T_5$ = Spent Mushroom Substrate + Biochar,

$T_6$ = Spent Mushroom Substrate + Poultry manure and

$T_7$ = Spent Mushroom Substrate + Vermicompost.

The chosen treatments were applied to the main field 20 days before transplanting the seedlings to ensure adequate decomposition, the growth of competitive microorganisms, and the development of the pathogen-suppressing abilities.

### Baseline soil physicochemical properties

Before the application of treatments, the soil at the experimental site of Sher-e-Bangla Agricultural University, Dhaka (AEZ-28, Madhupur Tract) was classified as silty loam, above flood level, upland topography, fair drainage system, dark olive-grey color, which contains approximately 31% sand, 41% silt, and 28% clay. The soil was moderately acidic, with a pH of 5.6, particle density 2.53 g/cc, organic matter content 0.78%, organic carbon 0.45%, and total nitrogen.06% while the available phosphorus and potassium were 20.9 ppm and 15.63 ppm, respectively. Available sulfur was recorded at 2.07 ppm. These baseline properties provided the initial soil condition against which the effects of the applied bio-organic amendments were evaluated.

### Preparation and characteristics of biochar

The biochar used in this study was produced from rice husk, a readily available agricultural by-product. The pyrolysis process was carried out in a muffle furnace at a temperature of 600°C for 4 hours under limited oxygen conditions to ensure proper carbonization. After pyrolysis, the biochar was cooled to room temperature, ground to pass through a 2 mm sieve,

and stored in airtight containers until application. The final product had a pH of approximately 8.5, indicating its alkaline nature. This biochar was selected based on its availability, environmental safety, and reported potential for suppressing soil-borne pathogens and improving soil health.

### Experimental design

The design of the study was laid out in a Randomized Complete Block Design (RCBD) with eight treatments and three replications. The experimental field area was 209 m² (19m x 11m), which was divided into three blocks. Each block had eight plots. As a result, three blocks had 24 plots. The size of each plot was (2.5 x 1.8) m. The space was 0.75 m kept between the blocks, and 0.50 m was kept between the plots. Each plot had 6 plants, and plant-to-plant distance was 75 cm (S1 Appendix).

### Data collection

On the basis of observable symptoms of fusarium wilt of brinjal, responses to the selected different treatments were noted at 25, 45, and 65 DAT (Days After Transplanting). The following parameters were considered for data collection- percent disease incidence, plant height, number of branches per plant, number of leaves per plant, number of fruits per plant, number of fruits per plot, fruit length, individual weight of fruits, yield per plant, yield per plot, and yield per ha. Yield per plant and disease incidence were calculated following the formula [15]:
   Where,

$$\textbf{Yield per plant} = \frac{\text{Total weight of all fruits per plot}}{\text{Total number of plants per plot}}$$

And

$$\textbf{Disease incidence } (\%) = \frac{\text{Infected number of plants  per plot}}{\text{Total number of plants per plot}} \times 100$$

### Isolation and identification of the pathogen

An infected stem was collected from the experimental field, and the stem was cut into small pieces (.05–1 cm) by a sterilized knife. Sterilization was done by dipping in 0.01% mercuric chloride ($HgCl_2$) for 2–3 minutes, and then placed on moistened blotter paper and then incubated in a humid chamber at $25 \pm 1°C$ for 7–10 days. When the fungus was grown then transferred to PDA media. The pathogen was then purified by the transfer of mycelium from the tip of the colony [16]. Finally, a compound microscope was used to study the morphological characteristics of the fungus.

### Pathogenicity test under pot culture

The soil was taken from the field. After that, a 0.4% formalin solution was completely mixed with soil at 7.0 L/m³ soil and kept under a polythene sheet for 48 hours to retain the gases within the soil for sterilization. The soil was then exposed to sunlight for seven days. After 7 days, the treated soil was ready to use. The soil was then placed in surface-sterilized pots with a diameter of 25 cm and a height of 30 cm, providing a total volume of around 15 liters. BT Brinjal 2 seedlings were raised in a plastic pot. Sterilized soil having fertilizers as per the package of practices was used for seedbed preparation. The inoculum of *Fusarium* sp. was prepared by culturing the pathogen on Potato Dextrose Agar (PDA) plates for 7 days at $25 \pm 2$ °C. Conidia were harvested by flooding the cultures with sterile distilled water and gently scraping the surface with a sterilized glass rod. The resulting suspension was filtered through double-layered muslin cloth to remove mycelial

fragments, and the concentration was adjusted to $10^6$ conidia/ml using a hemocytometer. Thirty-day-old seedlings of brinjal were treated with spore suspension of *Fusarium* sp. by the root dip method. The roots of 30-day-old brinjal seedlings were gently trimmed using sterilized scissors to slightly wound the root system, facilitating pathogen entry. The wounded roots were carefully washed to remove adhering soil particles and then immersed in 20 ml of a conidial suspension of *Fusarium* sp. ($10^6$ conidia/ml) for 10 minutes. Control plants were dipped in sterile tap water following the same procedure. After that, seedlings were placed into clean pots. The plants were watered regularly and observed for the appearance of wilt symptoms. Observations were done on wilt symptoms for up to 5 weeks. After three weeks of inoculation, symptoms were seen on the inoculated plant, and the pathogen was reisolated and compared with the original culture of *F. oxysporum* to satisfy Koch's postulates [16].

## Statistical analysis

The data were statistically analyzed by using the computer-based software Statistix 10 and by using analysis of variance (ANOVA) to find out the variation of results from experimental treatments. Treatment means were compared by LSD.

## Results

### Study on symptom and pathogen identification

Typical wilt symptoms were observed in the field, characterized by yellowing and wilting of the lower leaves, followed by progressive drooping and eventual death of the plant. A representative image of a wilt-affected brinjal plant is shown in (Fig 1). From the wilted plant, the pathogen *Fusarium* sp. was identified through microscopic examination based on symptomatology. On Potato Dextrose Agar (PDA) media, the fungus produced whitish to light pink mycelium, which gradually expanded into a light gray colony during sporulation. Microscopic analysis of the pure culture revealed the presence of 2–3 celled, slightly curved macroconidia, alongside single-celled microconidia (Fig 2).

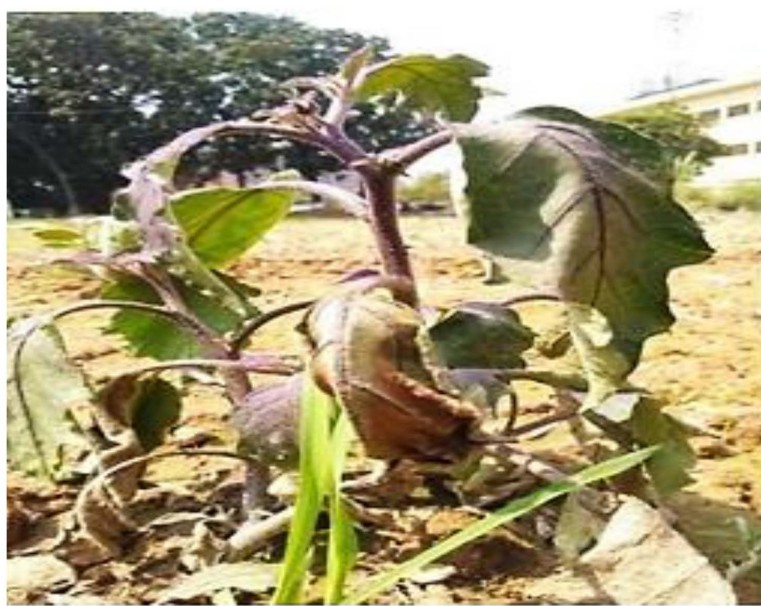

**Fig 1. An infected brinjal plant showing the typical symptom of Fusarium wilt in the field.**

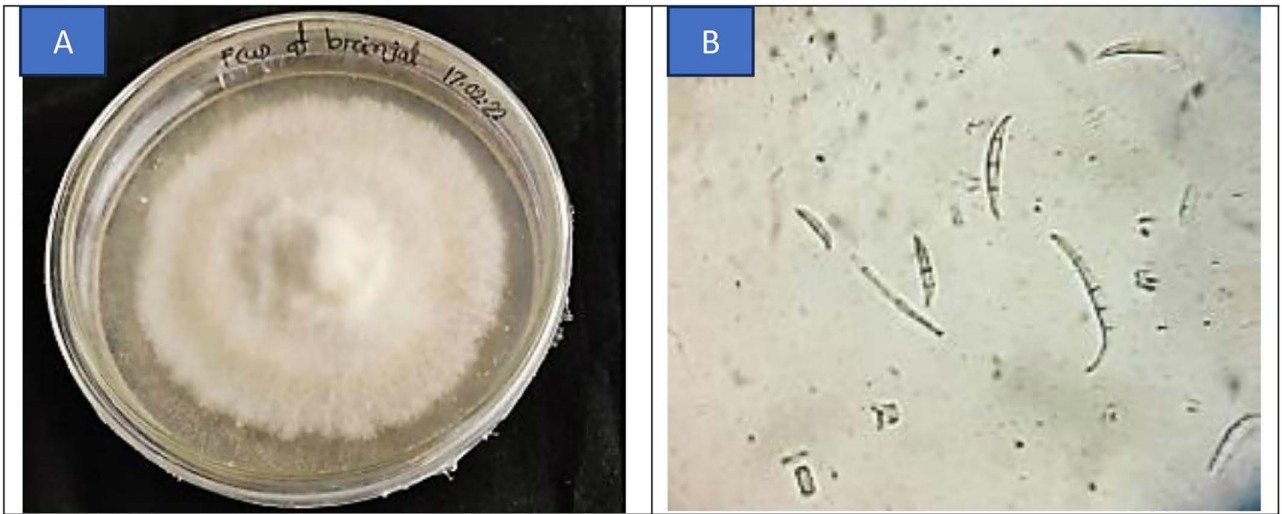

**Fig 2. Morphological study of *Fusarium* sp. A. Pure culture on PDA media, and B. Microscopic view under a compound microscope (40X).**

Pathogenicity was confirmed through typical wilting symptoms observed in brinjal plants inoculated with the isolated fungus. Re-isolation of *Fusarium* sp. from the stems of inoculated, wilt-affected plants completed Koch's postulates, verifying its role as the causal agent of brinjal wilt.

## Study on disease incidence

The influence of the selected bio-organic treatments on disease incidence was evaluated at 25, 45, and 65 Days After Transplanting (DAT). At 25 DAT, the untreated control ($T_0$) exhibited the highest disease incidence (44.4%), whereas the lowest incidence (5.55%) was recorded in $T_5$ (Spent Mushroom Substrate + Biochar). Similar trends were observed at 45 DAT and 65 DAT, where $T_0$ maintained the highest disease incidence (55.55% and 77.77%, respectively), while $T_5$ continued to show the most effective disease suppression (16.66% at both intervals).

Field observations confirmed that $T_5$ effectively controlled disease incidence across all time points, followed by $T_6$ (Spent Mushroom Substrate + Poultry Manure), highlighting the potent suppressive effect of organic amendments like spent mushroom substrate and biochar (Table 1).

**Table 1. Impact of different treatments on wilt incidence of brinjal at different days after transplanting (DAT).**

| Treatments | Disease Incidence (%) at 25 DAT | Disease Incidence (%) at 45 DAT | Disease Incidence (%) at 65 DAT |
|---|---|---|---|
| $T_0$ | 44.44 a | 55.55 a | 77.77 a |
| $T_1$ | 16.66 b-d | 22.22 bc | 33.33 bc |
| $T_2$ | 16.66 b-d | 38.87 ab | 49.99 b |
| $T_3$ | 27.77 a-c | 33.33 bc | 49.99 b |
| $T_4$ | 16.66 b-d | 38.87 ab | 44.44 b |
| $T_5$ | 5.55 d | 16.66 c | 16.66 c |
| $T_6$ | 11.10 cd | 27.77 bc | 27.77 bc |
| $T_7$ | 33.33 ab | 38.87 ab | 44.44 b |
| CV (%) | 58.83 | 35.99 | 35.97 |

## Study on the growth parameters of brinjal under different treatments

The treatments significantly influenced plant growth parameters, including plant height, the number of leaves per plant, and the number of branches per plant. The highest plant height (71.0 cm) was achieved in $T_5$, followed by $T_6$ (66.33 cm), while the lowest (50.66 cm) was observed in $T_0$ (Fig 3). For the number of leaves per plant, $T_5$ again exhibited the maximum count (111), significantly higher than $T_0$ (73.67), indicating improved vegetative growth. Similarly, the highest number of branches per plant (11.33) was recorded in $T_5$, statistically similar to $T_6$ (10.66), whereas the lowest (5.67) was found in $T_0$ (Fig 4).

## Study on yield and yield contributing attributes under different treatments

The effect of the selected treatments ($T_0$-$T_7$) against Fusarium wilt on yield and yield contributing characters of brinjal was studied, and significant variation was observed. The treatments not only suppressed disease incidence but also positively impacted yield-contributing attributes. $T_5$ demonstrated superior performance across all parameters, including the number of fruits per plant, the number of fruits per plot, individual fruit weight, and fruit length. These metrics were significantly higher compared to the untreated control ($T_0$), with $T_5$ yielding 12.27 fruits per plant, 73.66 fruits per plot, an average fruit weight of 92.52 g, and a fruit length of 9.01 cm (Table 2).

In terms of overall yield, $T_5$ produced the highest values (865 g/plant, 5.19 kg/plot, and 12.71 tons/ha), outperforming all other treatments, including the untreated control ($T_0$), which registered the lowest yield (277.67 g/plant, 1.66 kg/plot, and 4.08 tons/ha). $T_6$ also significantly improved, ranking second across all yield parameters (Table 3). These findings underscore the effectiveness of spent mushroom substrate combined with biochar ($T_5$) and poultry manure ($T_6$) in enhancing growth, reducing disease incidence, and improving yield in brinjal cultivation.

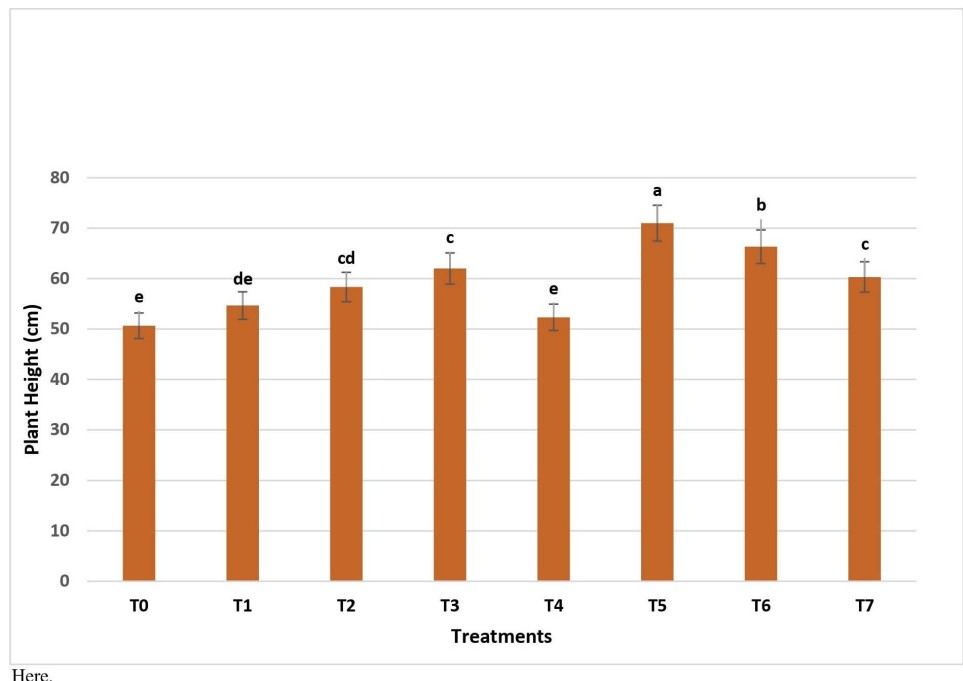

Here,

$T_0$ = Control; $T_1$ = Spent Mushroom Substrate (SMS); $T_2$ = Vermicompost; $T_3$ = Poultry manure; $T_4$ = Biochar; $T_5$ =Spent Mushroom Substrate + Biochar; $T_6$ = Spent Mushroom Substrate + Poultry manure; $T_7$ = Spent Mushroom Substrate + Vermicompost

**Fig 3. Impact of different treatments on the plant height of brinjal.** Here, T0 = Control; T1 = Spent Mushroom Substrate (SMS); T2 = Vermicompost; T3 = Poultry manure; T4 = Biochar; T5 = Spent Mushroom Substrate + Biochar; T6 = Spent Mushroom Substrate + Poultry manure; T7 = Spent Mushroom Substrate + Vermicompost.

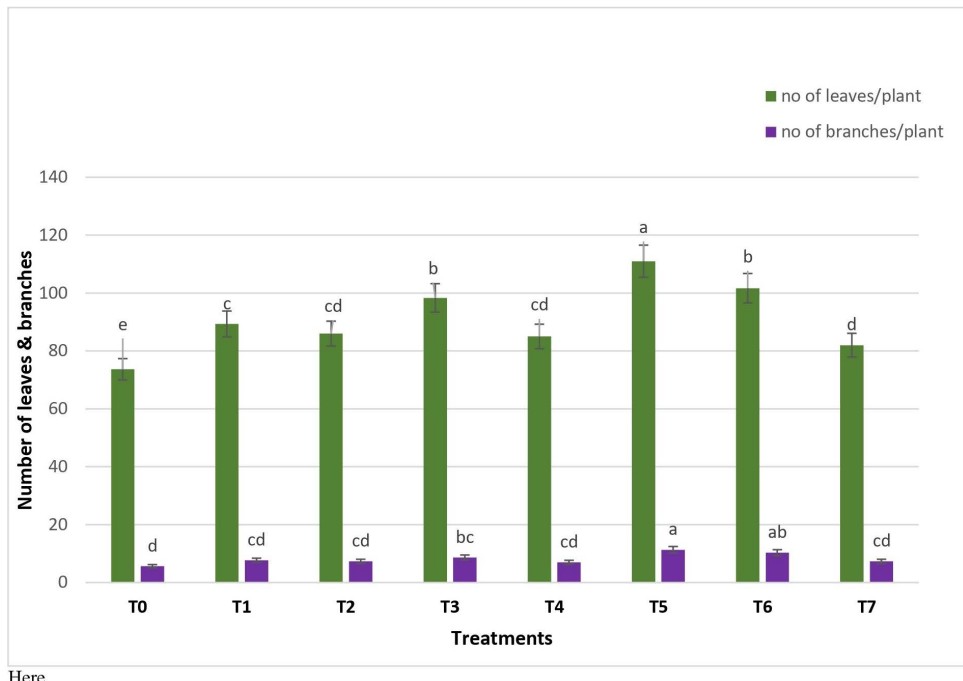

Here,

T_0 = Control; T_1 = Spent Mushroom Substrate (SMS); T_2 = Vermicompost; T_3 = Poultry manure; T_4 = Biochar; T_5 =Spent Mushroom Substrate + Biochar; T_6 = Spent Mushroom Substrate + Poultry manure; T_7 = Spent Mushroom Substrate + Vermicompost

**Fig 4. Impact of different treatments on the number of leaves and branches per plant of brinjal.** Here, T0 = Control; T1 = Spent Mushroom Substrate (SMS); T2 = Vermicompost; T3 = Poultry manure; T4 = Biochar; T5 = Spent Mushroom Substrate + Biochar; T6 = Spent Mushroom Substrate + Poultry manure; T7 = Spent Mushroom Substrate + Vermicompost.

**Table 2. Impact of different treatments on yield contributing characters.**

| Treatments | Number of fruit/plant | Number of fruit/plot | Individual weight of fruit (gm) | Fruit length (cm) |
|---|---|---|---|---|
| $T_0$ | 3.77 d | 22.66 d | 73.32 bc | 7.84 b |
| $T_1$ | 4.77 cd | 28.66 cd | 73.87 bc | 8.56 ab |
| $T_2$ | 7.77 b | 46.66 b | 66.69 c | 8.28 ab |
| $T_3$ | 6.77 bc | 40.66 bc | 73.14 bc | 7.80 b |
| $T_4$ | 8.38 b | 50.33 b | 81.73 b | 8.81 ab |
| $T_5$ | 12.27 a | 73.66 a | 92.52 a | 9.01 a |
| $T_6$ | 8.55 b | 51.33 b | 92.09 a | 9.11 a |
| $T_7$ | 4.72 cd | 28.33 cd | 76.44 bc | 8.21 ab |
| CV (%) | 19.35 | 19.33 | 7.12 | 7.16 |

## Discussion

### Pathogen identification

The identification of *Fusarium* sp. as the causal agent of wilt in brinjal aligns with previous findings by [8,17,18], where similar morphological characteristics and pathogenicity were observed. Consistent with these studies, the pathogen displayed whitish to light pink mycelium and the development of both 2 or 3 celled, slightly curved macroconidia and single-celled microconidia, confirming its identity through symptomatology and Koch's postulates.

**Table 3. Impact of the selected treatments on yield of brinjal against Fusarium wilt.**

| Treatments | Yield (g/plant) | Yield (kg/plot) | Yield (ton/ha) |
|---|---|---|---|
| $T_0$ | 277.67 d | 1.66 d | 4.08 d |
| $T_1$ | 353.67 cd | 2.12 cd | 5.19 cd |
| $T_2$ | 599.33 b | 3.59 b | 8.79 b |
| $T_3$ | 542.00 bc | 3.25 bc | 7.95 bc |
| $T_4$ | 691.00 ab | 4.14 ab | 10.12 ab |
| $T_5$ | 865.00 a | 5.19 a | 12.71 a |
| $T_6$ | 684.00 ab | 4.10 ab | 10.05 ab |
| $T_7$ | 364.67 cd | 2.18 cd | 5.33 |
| CV (%) | 21.97 | 21.97 | 22.17 |

## Disease incidence

The study demonstrated that treatments with Spent Mushroom Substrate (SMS) in combination with biochar ($T_5$) and poultry manure ($T_6$) significantly reduced disease incidence compared to the untreated control ($T_0$). The antifungal properties of SMS are well-documented, containing beneficial microorganisms like Trichoderma that exhibit antagonistic effects on soil-borne pathogens [19–21]. Furthermore, the integration of biochar in $T_5$ enhanced disease suppression by improving soil health and inducing systemic resistance in plants [22,23]. In 2020, Singh and Kumar reported that up to a 50% reduction in *Fusarium* growth due to biochar's inhibitory effects on chlamydospore development [24].

The combined application of SMS and poultry manure ($T_6$) also showed significant disease reduction due to the presence of organic amendments like poultry manure that suppress the viability of soil-borne plant diseases [25]. These outcomes underscore the potential of bio-organic treatments as eco-friendly alternatives to chemical control methods for managing Fusarium wilt in brinjal.

## Growth parameters, yield, and yield contributing attributes

The integration of SMS with biochar ($T_5$) and poultry manure ($T_6$) not only controlled disease incidence but also enhanced plant growth parameters and yield. This result is completely aligned with the report of [26] where they demonstrated that Biochar enhanced plant growth and activated plant resistance to various soil-borne pathogens. The observed increases in plant height, leaf count, branch number, and overall yield were indicative of improved soil fertility and microbial activity. This is consistent with the findings that demonstrated that SMS applications enrich soil biodiversity and enhance crop productivity [27]. The results suggest that incorporating SMS with other organic amendments could further improve soil properties and crop yields, supporting sustainable agriculture and reduced dependency on chemical inputs.

## Treatment-based variability in disease suppression and yield response

The observed variability in disease incidence and yield among the different treatments can be attributed to the distinct physicochemical and biological properties of the applied organic amendments. Treatments like SMS combined with biochar ($T_5$) and poultry manure ($T_6$) showed the most promising results due to their synergistic effects: SMS enriched the soil with beneficial antagonistic microbes such as *Trichoderma* spp., [28] while biochar improved soil aeration, moisture retention, and provided a conducive habitat for microbial colonization [13,29]. Poultry manure contributed readily available nutrients that supported plant vigor and enhanced resistance to pathogens [30]. In contrast, treatments like biochar alone ($T_2$), cow dung ($T_3$), or vermicompost ($T_4$) may have varied in their microbial activity, nutrient availability, decomposition rate, and soil pH influence, leading to less effective pathogen suppression or suboptimal plant growth. These differences

in the biochemical makeup, microbial load, and nutrient dynamics among the amendments directly influenced plant health, resistance mechanisms, and ultimately, the variability in growth performance and yield observed across treatments.

### Long-term implications of bio-organic amendments

*Fusarium* sp. is known for its persistent survival in soil due to its ability to produce chlamydospores, which can remain viable for 15 years or even more [31]. The long-term application of bio-organic amendments such as spent mushroom substrate (SMS), poultry manure, and biochar may contribute to sustained suppression of this pathogen by promoting a suppressive soil environment. These amendments enrich the soil with beneficial microorganisms such as *Trichoderma*, *Bacillus*, and actinomycetes, which are known to compete with or antagonize *F. oxysporum* [32]. Furthermore, biochar improves soil structure, water retention, and nutrient availability, potentially enhancing the resilience of the soil microbiome over time. While our study was limited to a single growing season, previous studies [33,34] suggest that consistent application of such amendments may progressively build disease-suppressive soils and reduce dependency on chemical control. Therefore, integrating bio-organic treatments in long-term management strategies could be a promising approach for sustainable control of Fusarium wilt and improvement of overall soil health.

## Conclusion

The findings of this study illustrate that the application of Spent Mushroom Substrate (SMS) in combination with biochar ($T_5$) and poultry manure ($T_6$) significantly reduces the incidence of Fusarium wilt in brinjal while simultaneously enhancing plant growth parameters and yield. The bio-organic treatments not only suppress the pathogen effectively but also enrich soil health by promoting microbial diversity and improving soil structure. Therefore, the use of SMS-based organic amendments presents a sustainable and eco-friendly alternative to conventional chemical methods for managing soil-borne diseases in brinjal cultivation, contributing to improved crop productivity and sustainable agriculture.

## Supporting information

**S1 Appendix. Field experiment layout for brinjal cultivation.**
(PDF)

## Acknowledgments

The author would like to express her heartfelt gratitude and appreciation to the Ministry of Science and Technology, Government of the People's Republic of Bangladesh, for bestowing the prestigious National Science and Technology (NST) Fellowship upon the author. This esteemed award has played a crucial role in supporting and enabling the successful completion of this study.

## Author contributions

**Conceptualization:** Jannatul Mauya, Khadija Akhter.

**Data curation:** Jannatul Mauya, Hasiba Kabir Mim, Raihan Ferdous.

**Formal analysis:** Jannatul Mauya, Marzia Rahman, Raihan Ferdous.

**Funding acquisition:** Khadija Akhter.

**Investigation:** Jannatul Mauya, Hasiba Kabir Mim.

**Methodology:** Jannatul Mauya.

**Project administration:** Khadija Akhter.

**Resources:** Jannatul Mauya, Marzia Rahman.

**Software:** Raihan Ferdous.

**Supervision:** Khadija Akhter.

**Validation:** Jannatul Mauya, Khadija Akhter, Hasiba Kabir Mim, Marzia Rahman, Raihan Ferdous.

**Visualization:** Jannatul Mauya, Marzia Rahman.

**Writing – original draft:** Jannatul Mauya, Raihan Ferdous.

**Writing – review & editing:** Hasiba Kabir Mim, Raihan Ferdous.

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
