## [Decision Letter · Decision Letter 0]

8 Jun 2025

Dear Dr. Ferdous,

Thank you for submitting your manuscript to PLOS ONE. After careful consideration, we feel that it has merit but does not fully meet PLOS ONE’s publication criteria as it currently stands. Therefore, we invite you to submit a revised version of the manuscript that addresses the points raised during the review process.

We look forward to receiving your revised manuscript.

Kind regards,

Ravinder Kumar, Ph.D.

Academic Editor

PLOS ONE

Journal Requirements:

Reviewers' comments:

Reviewer's Responses to Questions

**Comments to the Author**

1. Is the manuscript technically sound, and do the data support the conclusions?

Reviewer #1: Partly

Reviewer #2: Yes

Reviewer #3: Partly

2. Has the statistical analysis been performed appropriately and rigorously?

Reviewer #1: Yes

Reviewer #2: Yes

Reviewer #3: Yes

3. Have the authors made all data underlying the findings in their manuscript fully available?

Reviewer #1: Yes

Reviewer #2: Yes

Reviewer #3: Yes

4. Is the manuscript presented in an intelligible fashion and written in standard English?

Reviewer #1: Yes

Reviewer #2: Yes

Reviewer #3: No

Reviewer #1: Using organic amendments in soil can have positive effect on plants' yields and improve plant health by increasing soil microbiome.

The manuscript (MS) conforms to the journal's requirements.

Abstract

line 17: The authors state that the pathogen was identified by morphological and pathogenicity tests. However there are several very similar Fusarium species which can infect S. melongena. I'd suggest to do at least a PCR test for tef1 gene and then sequence the specific product to have a molecular identification of the pathogen. OR the authors could have used an already molecularly identified isolate from a culture collection.

It is also not clear if the experiment was done once or it was repeated and if there were repetitions of the treatment. This important information is in the Materials and Methods section, please add it to the abstract, because it makes the Author's research more valuable.

Introduction

line 32, Please make Solanaceae italic.

line 35: MT, please write out the first time metric ton and then use the abbreviation in parentheses (MT), later authors can use only the abbreviation.

lines 42-44: Please rephrase the sentence and note that there are several Fusarium species which can infect brinjal, but the most important of these species is the most devastating, Fusarium oxysporum f. sp. melongenae.

lines 46-47: please add that Fusarium oxysporum survives in the soil with its chlamydospores.

line 48: measures (use the plural of measure)

line 54: Bio-control agents are required to go through several official test in Europe, USA and Australia among other countries. Instead of bio-control measure I would use the terms "organic soil additives" or "organic byproducts". I would use use "poultry manure" instead of "poultry refuse".

line 56: please explain what is SMC

line 57: please explain what is SMS (like in the Abstract the Authors did)

line 60: use plural of earthworm

Materials used

lines 74-76: delete it, it is redundant.

line 106: I feel that 6 plants are not enough to make the statistics reliable. The more plants one has in a treatment the more reliable the results are.

line 123: add "moistened" blotting paper.

line 124: add "in a humid chamber".

line 132: what was the volume of the pots used?

line 136: how were the roots damaged? Did the authors tried just dipping the seedlings in the spore suspension?

Results

lines 149-158: In scientific publishing it is expected that the Authors carry out pathogen identification with 2 different methods, one method should result in a molecular evidence, like sequencing a genetic barcode region or housekeeping gene. Without molecular evidence one cannot reliably state that a pathogen is identified and causes the disease. For Fusarium species the tef1 gene is used for identification. This gene should be used to identify the pathogen before and after carrying out the Koch's postulates. Scientific rigor also requires that the entire trial is repeated at least twice in consecutive growing seasons or at different locations.

What I'm missing is quality photographs of the wilted plants to show the effectiveness of the different treatments. Because all the plants were potted it would be easy for the Authors to collect 1 demonstrative example from each treatment, put them side by side and take a picture. This visualization would greatly enhance the value of the publication. There are no symptom photographs, just tables graphs and an image from a culture and macro-conidia.

I also would expect tho have this tiral setup repeated at a different location or different year, because weather changes from year to year and from location to location. In science one cannot draw reliable conclusion from one trial from one location.

This trial has shown great potential for the use of bio-organic matters as soil amendments to combat Fusarium wilt of Solanum melongena. However it appears that the results and conclusions are not supported well enough with data in this current form of the MS, so I suggest a major revision and expansion of the results section from data from a different location OR from at least 2 growing seasons.

Reviewer #2: The manuscript titled “Optimizing brinjal (Solanum melongena L.) health and yield through bio-organic amendments against Fusarium wilt” was submitted for consideration in PLoS One. The manuscript demonstrates the study on an important issue in sustainable brinjal cultivation by exploring eco-friendly alternatives to chemical fungicides. The pathogen was accurately identified using appropriate morphological and pathogenicity tests. Significant reductions in disease incidence and improvements in growth and yield were demonstrated with bio-organic amendments, particularly the combination of spent mushroom substrate with biochar or poultry manure. The results are supported by clear data and sound statistical analysis. The manuscript is generally well-written, and the findings are placed thoughtfully within the context of existing research, highlighting the practical relevance of the work for sustainable agriculture. However, it is not suitable for publication in its present form. The following improvements should be made for further consideration.

1. The assessment of microbial diversity should be expanded by including detailed methods or data on soil microbial communities to strengthen related claims.

2. A control treatment with a chemical fungicide should be included or discussed to provide a benchmark for comparing the efficacy of bio-organic amendments.

3. The characteristics of the biochar used, such as source material and pyrolysis conditions, should be detailed to enhance reproducibility and understanding of treatment effects.

4. The variability observed in disease incidence and yield data, indicated by high coefficients of variation, should be addressed with discussion of potential causes and management approaches.

5. Soil physicochemical analyses, including nutrient status and pH before and after treatments, should be provided to support conclusions about soil health improvements.

6. The long-term implications of the treatments on disease suppression and soil health should be discussed, considering the persistent nature of Fusarium oxysporum.

7. The quality and clarity of figures and legends should be improved by adding detailed descriptions and explaining all symbols and abbreviations.

8. Minor grammatical and typographical errors should be corrected through careful proofreading for polished presentation.

9. As the results demonstrate that the treatment with biochar outperformed the other treatments in different characters. The authors must give credit to and cite the recent literature on the application biochar for disease resistance in different crops, for instance,, the review by Arshad (2024) (10.55627/pbiotech.002.01.0972), Arshad et al. (2020, https://doi.org/10.17957/IJAB/15.1522) and (2021, https://doi.org/ 10.1007/s10343-021-00580-4) are highly relevant to this research and should be incorporated in the discussion.

10. Similarly, recently, Fayyaz et al. have reported different biochars for nematode management in tomato (10.21162/PAKJAS/24.30). These reports must be cited in this review of literature.

Reviewer #3: The manuscript addresses an important aspect of sustainable agriculture: the use of bio-organic amendments to manage soil-borne diseases like Fusarium wilt in brinjal (Solanum melongena L.). The topic is relevant, particularly in the context of integrated disease management and organic farming practices. However, the manuscript exhibits several limitations in scientific depth, methodological clarity, and data interpretation.

The abstract provides a broad summary of the study's intent but lacks numerical data and statistical validation of findings. It overstates the results without sufficient context or limitations.

The introduction is overly general and lacks citations from recent, high-impact literature. No clear hypothesis is defined. Rationale behind the selection of bio-organic treatments is poorly developed.

In methodology experimental design lacks clarity like randomization, replication, and plot size are not clearly mentioned. Similarly, no information on soil physico-chemical properties before treatment application is presented.

Also the absence of details on Fusarium wilt inoculum preparation and application undermines the disease pressure standardization and data collection frequency as well as disease rating scale is not sufficiently described.

The discussion lacks depth and critical analysis. Several repetitive and descriptive statements rather than analytical in the discussion. Also add discussion of limitations, which is essential for transparency. Similarly, author failed to explore the mechanism behind the suppression of Fusarium wilt.

Also, the manuscript contains multiple typographical errors, grammatical mistakes, and inconsistent formatting. Scientific names are inconsistently italicized. Units and abbreviations are not standardized throughout.

Although, the manuscript holds potential but requires substantial improvement to meet publication standards, so my recommendation is Major Revision Required

**Do you want your identity to be public for this peer review?** For information about this choice, including consent withdrawal, please see our Privacy Policy

Reviewer #1: No

Reviewer #2: No

Reviewer #3: **Yes: ** Raees Ahmed

---

## [Author Response · Author response to Decision Letter 1]

9 Aug 2025

We attached a separate file to response the reviewers. Please follow the file.

---

## [Decision Letter · Decision Letter 1]

27 Aug 2025

Dear Dr. Ferdous,

Thank you for submitting your manuscript to PLOS ONE. After careful consideration, we feel that it has merit but does not fully meet PLOS ONE’s publication criteria as it currently stands. Therefore, we invite you to submit a revised version of the manuscript that addresses the points raised during the review process.

We look forward to receiving your revised manuscript.

Kind regards,

Ravinder Kumar, Ph.D.

Academic Editor

PLOS ONE

Journal Requirements:

Reviewers' comments:

Reviewer's Responses to Questions

**Comments to the Author**

Reviewer #1: (No Response)

Reviewer #2: (No Response)

2. Is the manuscript technically sound, and do the data support the conclusions?

Reviewer #1: Partly

Reviewer #2: Yes

3. Has the statistical analysis been performed appropriately and rigorously?

Reviewer #1: I Don't Know

Reviewer #2: Yes

4. Have the authors made all data underlying the findings in their manuscript fully available?

Reviewer #1: No

Reviewer #2: Yes

5. Is the manuscript presented in an intelligible fashion and written in standard English?

Reviewer #1: Yes

Reviewer #2: Yes

Reviewer #1: Authors have addressed most of the comments I made in the first round of review. However I still feel that this MS does not meet scientific rigor.

Line 155: "@ 200ml/cft" - I'd kindly ask the authors not to use @, instead write out "at" and instead of using cft (= cubic feet?), I strongly recommend to use SI units as recommended in the Journal's submission guidelines.

In line 160-161 the authors state the used Fusarium oxysporum f. sp. melongenae, but in their response they state the pathogen was identified my classical methods, and the original cultures are not available for further molecular testing. Identifying a pathogen by only classical methods does not meet scientific rigor not allows reproducibility.

Line 169: The authors should state what was the volume of the conidial suspension. It is also not clear weather the roots of the seedlings were washed from soil or they carried out the dip method with soil. If it was carried out with soil the soil absorbs part of the conidial suspension it was dipped in, this is why the volume of the conidium suspension is important. More absorbed volume menas more inoculum per plant.

line 279-282: "The authors reported that the application of SMS increased the diversity of fungi, including Tremellomycetes and Pezizomycetes for the SMS additive, while the levels of Mortierellomycetes, Pezizomycetes, and Leotiomycetes increased after the addition of poultry manure." This sentence does not make sense and should be deleted. The authors did not test the microbial community of the plots before or after the trial, and have no data to substantiate this statement.

Reviewer #2: The authors have significantly improved the manuscript. For strengthening the introduction and discussion part they must add the suggested citations. As the results demonstrate that the treatment with biochar outperformed the other treatments in different characters. The authors must give credit to and cite the recent literature on the

application biochar for disease resistance in different crops, for instance,, the review by Arshad (2024)

(10.55627/pbiotech.002.01.0972) where the author has demonstrated the role and mechanisms of biochar for enhancing resistance to pathogens and insects.

**Do you want your identity to be public for this peer review?** For information about this choice, including consent withdrawal, please see our Privacy Policy

Reviewer #1: No

Reviewer #2: No

---

## [Author Response · Author response to Decision Letter 2]

7 Sep 2025

Response to reviewers is uploaded as a pdf file.

---

## [Decision Letter · Decision Letter 2]

24 Sep 2025

Optimizing brinjal (Solanum melongena L.) health and yield through bio-organic amendments against Fusarium wilt

PONE-D-25-25430R2

Dear Dr. Ferdous,

We’re pleased to inform you that your manuscript has been judged scientifically suitable for publication and will be formally accepted for publication once it meets all outstanding technical requirements.

Kind regards,

Ravinder Kumar, Ph.D.

Academic Editor

PLOS ONE

Reviewers' comments:

Reviewer's Responses to Questions

**Comments to the Author**

Reviewer #2: All comments have been addressed

2. Is the manuscript technically sound, and do the data support the conclusions?

Reviewer #2: Yes

3. Has the statistical analysis been performed appropriately and rigorously?

Reviewer #2: Yes

4. Have the authors made all data underlying the findings in their manuscript fully available?

Reviewer #2: Yes

5. Is the manuscript presented in an intelligible fashion and written in standard English?

Reviewer #2: Yes

Reviewer #2: The authors have addressed the comments, have made necessary corrections and it should be accepted.

Regards,

**Do you want your identity to be public for this peer review?** For information about this choice, including consent withdrawal, please see our Privacy Policy

Reviewer #2: No

---

## [Editor Report · Acceptance letter]

PONE-D-25-25430R2

PLOS ONE

Dear Dr. Ferdous,

I'm pleased to inform you that your manuscript has been deemed suitable for publication in PLOS ONE. Congratulations! Your manuscript is now being handed over to our production team.

Kind regards,

on behalf of

Dr. Ravinder Kumar

Academic Editor

PLOS ONE